# Structural Health Monitoring in Long-Span Steel Structures Based on the BeiDou Navigation Satellite System

**DOI:** 10.3390/s23135959

**Published:** 2023-06-27

**Authors:** Hui Gao, Baoxin Jia, Guochuan Wang, Tong Zhang, Pei Dang, Donglin Wang

**Affiliations:** 1School of Civil Engineering, Liaoning Technical University, Fuxin 123000, China; zhangtong@lntu.edu.cn (T.Z.);; 2Liaoning Key Laboratory of Mine Subsidence Disaster Prevention and Control, Liaoning Technical University, Fuxin 123000, China; 3Second Engineering Co., Ltd., China Communications Construction Group, Nanchang 330000, China; 4School of Economics and Management, Tianjin Chenjian University, Tianjin 300392, China

**Keywords:** Global Navigation Satellite System, BeiDou navigation satellite system, structural health monitoring, long-span steel structures, Tianhan System, combination of techniques

## Abstract

The BeiDou navigation satellite system (BDS) provides precise positioning, navigation, and timing (PNT) services in the Asia-Pacific Region, but the BDS-based structural health monitoring (SHM) approach (SHM) is rarely studied, especially in civil engineering. Moreover, how BDS can be applied to complete the tasks of SHM in a real project is also not fully investigated, especially working in conjunction with other techniques. This study aims to propose a BDS-based approach for SHM in civil engineering. The performance of the proposed approach is investigated through a case study—the Tianhan Grand Theater (TGT). A specific Tianhan system corresponding to BDS is proposed to complete the SHM tasks of TGT. Based on the collected data, the trusses with maximum displacement and stress are found by BDS to evaluate structural health in the construction stage. The results show that the maximum displacement and stress have certain safety reserves and meet the requirements of the specifications and regulations. Thus, BDS can satisfactorily complete the tasks of SHM for Long-span steel structures. This study gives a clear view to engineers and researchers that how to apply BDS in structural construction and provides a valuable real case for evaluating the performance of BDS in SHM.

## 1. Introduction

Long-span steel structures have been quite useful in the modern world over recent years because of their advantages, such as the esthetical appearance, lightweight, low-cost, flexible architecture, good stability, and high strength [1]. However, as they can cover large areas of uninterrupted spans, which is one of the most attractive advantages, the deformation problems of long-span steel structures are always a concern for engineers, designers, and researchers. Unfortunately, the deformation problems cannot be solved well by traditional deformation monitoring technologies as high precision is required for detecting small and slow displacements [2]. To be specific, during construction, certain deviations can exist in the process of positioning, installation, and welding of steel components. Moreover, during the temporary support removal process, the structural load changes greatly, which can lead to deformation in different areas of the steel structures [3]. For traditional deformation monitoring technologies, the most commonly used instruments are measuring robots; however, the cost of measuring robots is high and they cannot monitor all parts of the steel structures [3]. Compared to traditional deformation monitoring technologies, the Global Navigation Satellite System (GNSS) positioning technology is dominating the current deformation monitoring field because of its advantages, such as dispensing with intervisible measuring stations, all-climate working, direct access to the 3D coordinates of stations, and high degree of automaticity [4]. GNSS has been one of the best choices used in all types of structural deformation monitoring in recent years because of its irreplaceably automated, all-weather, continuous, and highly accurate measurement services [5]. Most importantly, the precision of GNSS can fulfill the requirements of most structural health monitoring (SMH) tasks for types of structures [6,7].

Currently, GPS is the dominating GNSS system in the deformation monitoring field, and nearly all of the research conducted has been based on GPS, which consists of 32 satellites at the amplitude and provides worldwide coverage and geo-positioning [5]. GPS has been assessed to be capable of achieving high precision and feasibility for engineering applications by many studies [7,8,9,10,11]. However, to date, there are still many problems to solve with GPS to provide reliable monitoring information for long-span steel structures [12]. For instance, due to obstructions, the observation environment on the long-span steel structures is not always good enough for GPS observation [12,13,14,15]. Most importantly, GPS will not be available for countries that cannot obtain permission from the US.

Fortunately, except for the GPS, the BeiDou navigation satellite system (BDS) also is providing precise positioning, navigation, and timing (PNT) services in the Asia-Pacific Region [16,17]. BDS is a global satellite navigation system that is independently developed, deployed, and fully operated by China in 2020 [18,19]. There are 30 satellites in the BDS constellation, including geostationary orbit (GEO), inclined geosynchronous orbit (IGSO), and medium earth orbit (MEO). Currently, BDS is providing comparable satellite visibility and availability for positioning, navigation, and timing (PNT) to all types of users at any time, all-weather, and anywhere in the world since 2020 than GPS because of its better signal quality, similar code division multiple access (CDMA) modes, and different constellations [18,19,20].

Many studies have assessed the accuracy of BDS for positioning and navigation, and it is proved that BDS has all the advantages that GPS has and can provide similar or even better precision and stability for users [5,21,22,23,24,25,26]. The radial precision of the BDS satellite orbit determination is better than 10 cm, which is comparable to those of GPS [19]. The accuracy of BDS carrier phase differential positioning is better than 1 cm for a very short baseline and 3 cm for a short baseline, which is on the same level as that of GPS; the radial precision of the BDS satellite orbit determination is better than 10 cm, and the accuracy of BDS code differential positioning is better than 2.5 m [12]. Moreover, in the midlatitude of China, BDS can achieve an accuracy of 1 mm in the horizontal component and 2–3 mm in the vertical component with the post-processed mode [5]. These performances confirm that BDS positioning precision and navigation applications are comparable to or even better than those of GPS [16,27]. Moreover, if GPS and BDS are combined, the positional and navigation precision can be further enhanced [20,28].

Structural health monitoring (SHM) refers to the strategy and process of damage identification and characterization of engineering structures. With the progress of society and the development of the economy, a large number of engineering constructions are carried out everywhere, and hence SHM is becoming more and more significant in civil engineering because of increasing structural damage issues [29,30]. Thus, several approaches for conducting SHM were addressed by scholars in recent decades. Tiachacht et al. [31] developed a Modified Cornwell Indicator (MCI) approach for quantifying the damaged elements in frame structures, and the results showed that MCI can predict the presence, location, and severity to quantify damaged elements. Ghannadi et al. [32] used Grey Wolf Optimization (GWO) approach for SHM of skeletal structures. Rao and Patel [33] proposed a Teaching-Learning-based optimization (TLBO) algorithm to simulate the teaching-learning process of the classroom, and provided that the proposed algorithm can be applied to various optimization problems of the industrial environment. Yang and Deb [34,35] proposed a new metaheuristic algorithm, which is called Cuckoo Search (CS), for solving optimization problems, and found that the optimal solutions obtained by CS are far better than the best solutions obtained by an efficient particle swarm optimizer. Benaissa et al. [36] proposed a new metaheuristic algorithm, which is called as YUKI Algorithm (YA), with a search space reduction capability to evaluate defective structures. The results showed that YA can provide more accurate and faster results than previous algorithms such as TLBO, CS, and GWO. Khatir et al. [30] investigated the performance of the Whale Optimization Algorithm (WOA) technique for model updating and damage identification via two cases, and the results show that WOA has better accuracy than the compared techniques to accurately and efficiently estimate both damage location and severity in frame structures. Slimani et al. [37] suggested an approach for vibration sensor placement in Carbon Fibre Reinforced Polymer (CFRP) composite structures, and the results showed that the proposed application can explain the effect of sensor placement. Kahouadji et al. [38] also proposed an approach based on Local Frequency Change Ratio (LFCR) as a damage indicator and optimization techniques to conduct damage identification. The results showed that the LFCR can detect and locate the damage precisely and the presented optimization techniques can define its severity accurately. It can be found that the previous approaches can effectively carry out structural damage assessment, but a BDS-based approach is rarely studied and paid much attention to. Moreover, although the accuracy of BDS for positioning and navigation has been assessed by studies, the performance of BDS-based applications in deformation monitoring is rarely studied [5,12], and the deformation monitoring accuracy of BDS in construction processes has also not yet been fully analyzed and assessed. Moreover, how BDS is used in a real project for structural health monitoring (SHM) is also rarely studied, especially for long-span steel structures. Furthermore, most of the previous studies only investigate BDS but neglect the key role played by other closely related techniques, such as BIM, FEA, and ZigBee. Actually, BDS should be used in conjunction with other techniques to carry out the SHM for structural construction and operation. However, attention is rarely paid to how to smoothly apply BDS with other techniques, which can be quite significant for engineers and researchers. Moreover, how to specifically apply BDS in a real project, such as the selection of hardware, the production of the corresponding applications (apps), and the site layout, are also rarely investigated. Thus, it is necessary to use a case study as a demonstration to investigate the performance of BDS in SHM and how to apply BDS smoothly. 

This study aims to propose a BDS-based approach to carry out structural health monitoring in civil engineering. The performance of the proposed approach in structural health monitoring is also assessed. To do this, a famous project—Tianhan Grand Theater—is chosen as the case study to demonstrate how the BDS-based approach works in a real project with the support of other techniques, how to identify the critical components, and what satisfactory results can be achieved. The first objective of this study is to introduce the creation processes of the Tianhan System, which is used for the BDS-based SHM approach. The second objective is to assess the performance of the proposed approach based on the collected data and evaluated results. This study can give a clear view to engineers and researchers on how to apply BDS for long-span steel structures and provide a valuable BDS-based approach in the SHM area to supplement the existing research gap. It can be predicted that this study will be helpful in solving structural damage-related issues in real engineering applications.

## 2. Project Background and Tianhan System

### 2.1. Project Background

Tianhan Grand Theater (TGT) is designed and constructed as one of the core buildings in the city center of Hanzhong (as shown in Figure 1), which gathers the municipal large-scale comprehensive performance theater, multifunctional small theater, senior citizen activity center, conference center, and many other supporting service rooms. It will be the landmark building of Hanzhong City and can provide comprehensive cultural services for the people. Tianhan Grand Theater is planned to cover a total land area of 6.56 hectares, with a building height of 33.75 m and a total construction area of 69,994 square meters, including 46,691 and 23,303 square meters above and below ground, respectively. The north building has four floors above ground, while the south building has five floors, both of which have one floor underground. The main body of the steel structure adopts the Mobius ring shape, which is unique and novel, and hence can be called the Mobius ring steel structure (MRSS). The MRSS is separated into two layers inside and outside, respectively, constructed with trapezoidal-section bending and twisting steel box members of different sizes, which are cross-woven with each other.

Located outside the main concrete structure, the main body of the steel structure consists of 130 trusses of rigid frames in the form of a ‘radial arrangement + annular connecting rod + annular aisle rectangular truss + column support’ as a Mobius ring. The plane shape of the MRSS is approximately circular; the maximum projection length is about 178 m, the maximum height of the structure is 33.75 m, and the maximum span is about 42 m. The section forms of the structural components are box and round tube members. The amount of used steel is about 3600 tons, and the material is mainly Q355B steel. The composite truss multi-bars are conjunct at the main entrances using cast steel members, one end of which is fixedly connected with the annular walkway and the other end is hinged with the ground, as shown in Figure 2a,b.

### 2.2. The BDS-Based SHM System

The construction of the large-scale spatial steel structure is extremely complex, as shown in Figure 3, and the load is changeable. It is a dynamic and instantaneous change process of structure, since the construction progress is gradually promoted, and the main structural components are gradually increased. The temperature change, mutual restraint of components and structural shapes can lead to a change in the overall structural stiffness, stability, and other parameters, which can significantly affect the structural health and need to be monitored. For example, the National Stadium, the National Grand Theater, and the Water Cube have all been monitored for stress and displacement deformation to ensure structural health. Thus, it is also very necessary to monitor the stress and displacement deformation of the MRSS for this project. The monitoring of the construction stage is mainly based on the actual construction processes, and the parameters and data obtained on the site are used to carry out real-time theoretical analysis and structural inspecting calculations of the structure. For each construction stage, the structural analysis and calculation results are used to determine the error in construction, and the early warning system is adopted to carry out safety degree evaluation and disaster warning for the construction onsite. The monitoring can provide a decision-making technical basis for the construction process, which correctly assists the decision-maker to ensure that the structural linearity and stress state meet the design requirements. With this in mind, a structural health monitoring system (Tianhan System) is proposed for Tianhan Grand Theater based on the Beidou GNSS system and ZigBee wireless communication technology, which not only has a high degree of automation but also can realize real-time millimeter-level three-dimensional deformation monitoring. The system aims to monitor the stress and displacement deformation of key parts of the MRSS to ensure that the stress and displacement deformations of the key components are in a reasonable range to guarantee the safety and reliability of the whole construction process, and to put forward a study on intelligent tracking simulation at the construction and operation stages.

### 2.3. BIM

As a part of the Tianhan System, the Building Information Modeling (BIM) technique works in conjunction with BDS to achieve the objective of SHM. The BIM software of TeklaStructures is used by this project to build a completable building information model of the MRSS’s construction, in which details of the whole structure can be displayed completely and accurately, providing an important reference and basis for the subsequent component construction processing. The BIM work of this project is complex, which includes several procedures such as node fine modeling construction, modeling of steel concrete beam–column joint, fine high-strength bolts modeling construction, and modeling of complex steel joints, which are out of the scope of this study and will not be introduced. Attention is mainly paid to the overview of the MRSS model, which is closely related to the application of BDS. Figure 4 shows a part of the finished modeling of the outer curtain wall steel structure, and Figure 5 gives an overview of the whole modeling of the Mobius ring steel structure.

The MRSS has complex joints, three-dimensional variable cross-section structures, and different rod sizes. It is difficult to install steel components at high altitudes and accurately calculate the tilt angles. Moreover, positioning and lofting the components in space measurements onsite are also tough tasks. At this time, it is necessary to adopt the combination of BDS and BIM three-dimensional model for spatial positioning measurement, that is, to determine the coordinate points of the steel structural model in the Tekla structure based on the site lay-off point position, and to export the coordinates of each point position (as shown in Figure 6) to BDS. During assembly, after alignment with the origin, the position of each rod and node is controlled by the exported coordinates with the help of BDS. After hoisting in place, measurement and correction are carried out according to the three-dimensional points of the Tekla Structure model to ensure the accuracy of the high-altitude installation, which can ensure the construction accuracy, reduce adjustment and rework time, shorten the installation period, and reduce labor and material costs as far as possible. Figure 7 shows how to combine the BDS with the BIM three-dimensional model for onsite installation and positioning measurement.

### 2.4. Numerical Calculation Model

BDS also requires the support of other techniques to previously identify the most vulnerable parts of the MRSS for performing the most cost-effective monitoring. As introduced above, the steel structure’s inner and outer ring trusses and the rigid frame are mounted with the support frame, and during the construction process, the force assumed by the original design of the structure can be changed. Therefore, in order to guide the safe implementation of the construction scheme of the project, the construction process of the structure needs to be simulated and analyzed. Using 3D3SDesign2020 software, the numerical calculation model of the MRSS was established, and the construction and unloading processes of the structure were simulated and analyzed. Figure 8a illustrates the last step of lifting as the example, and Figure 8b shows the last step of unloading. The eight-node linear reducing integral hexahedron elements (C3D8R elements) were used to model steel frame members. The material properties of the steel members were modeled with the mechanical properties of the Q355B steel recorded in the standard of High strength low alloy structural steels (GB_T1591-2018) [39], with a yield stress of 355 MPa, Poisson’s ratio of 0.3, and Young’s modulus of 2.1 GPa. The stress vs strain curve was converted to the true stress vs true strain curve for the material property definitions.

According to the simulated model, the critical trusses and joints were found and specifically monitored by the Beidou System.

### 2.5. ZigBee and Hardware Preparation

Besides the support of BIM and numerical calculation, specific hardware and software systems are also required by Tianhan System for the objective of SHM. In consideration of this, the ZigBee technology is applied in this project. The Tianhan Beidou–Zigbee data acquisition and transmission system, which can be shortened to the Tianhan System, consists of four parts: ZigBee sensing terminal node, ZigBee coordination module, Beidou module, and embedded Linux upper computer. The ZigBee sensing terminal node collects data from the sensor and transmits the data to the ZigBee coordination module through the ZigBee star network. Then, the ZigBee coordination module compresses packages of data and appends the Beidou protocol. At last, the Beidou module transmits the data to the upper computer for real-time display. Figure 9 shows the overall architecture of the data acquisition and transmission system based on BDS and ZigBee. For the Beidou module, it includes the Beidou command computer and the Beidou user computer. The Beidou command computer is connected to the upper computer and receives data from the subordinate user computer. The user computer is connected to the ZigBee coordination module and receives data from the ZigBee coordination module. There are plenty of user computers that are distributed in a large area for conveniently displaying multiple data in just one upper computer.

The ZigBee terminal node is responsible for connecting with the sensor device, receiving the analog signals of the sensor, and converting and transmitting digital signals to the ZigBee coordination node. In addition to the basic tasks of building a network and allowing other nodes to attend and exit, the received data also need to be pre-processed, packaged, compressed, and then sent to the Beidou module after adding the Beidou protocol. Since the minimum communication interval of Beidou communication is 1 min and the amount of data sent in a single time is relatively small, the CC2531 chip can fully meet the requirements without using other advanced chips. Its hardware block diagram is shown in Figure 10. In addition, the upper computer is connected to the Beidou controller through the UART serial port, and the upper computer chip is Samsung’s S3C2440 with a 7-inch touchscreen.

### 2.6. Tianhan App

The Tianhan System also requires its own app for operation, and hence an app, whose name is Dynamic Monitoring System of Large-scale Space Steel Structure Construction in Tianhan Grand Theater (shortened as Tianhan App), is proposed as a Beidou real-time data stream high-precision processing application. Tianhan App includes three programs: GNSS real-time data processing back-end program, data transmission program, and real-time data visualization monitoring program. It can obtain the position information, satellite visibility, calculation reliability, and PDOP value of the components in the real-time installation process. It can also show the distance and direction information between the component and the position to be installed, and carry out a real-time three-dimensional display of the interface, which is convenient for installation personnel.

The Tianhan App supports multi-system integration and multi-frequency band calculation. It also supports data communication, real-time data decoding, and real-time location settlement of various communication protocols such as Ntrip protocol and TCP/IP protocol. The app can decode the received real-time data stream and identify the position of components, and then use the calculated result for real-time position monitoring and display for supporting the application of component welding and deformation insurance. For the real-time data visualization monitoring program, the overall model of Tianhan Grand Theater can be displayed within the app, as shown in Figure 11. The app includes three functional modules: the construction module, the deformation monitoring module, and the stress monitoring module, which can meet the requirement of real-time dynamic construction and monitoring of steel structures in Tianhan Grand Theater.

The construction module can obtain the position information, satellite visibility, calculation reliability, and PDOP value of the components in the real-time construction process. After the completion of construction, the deformation and stress monitoring shall be carried out, and the other two modules shall be used. The deformation and stress monitoring modules are shown in Figure 12 and Figure 13, respectively. When the stress or deformation value is greater than the threshold, the app indicates that the component is faulty. More specifically, in the stress/displacement change warning window, a message can be found to specify the component that exceeds the threshold, which can help the installation personnel to discover the fault promptly.

## 3. Application of the Tianhan System

### 3.1. Overview

The BDS-based approach is applied via the Tianhan system to monitor structural health. To summarize briefly, the Tianhan system is created by using techniques of BIM, numerical simulation, ZigBee wireless, and BDS. The BIM technique is used to create the 3D model and provide the coordinates of each point position for spatial positioning measurement. The numerical simulation technique is used to perform the finite element analysis to find the critical trusses and joints that require special monitoring. The ZigBee wireless technique is used to connect the sensors and the BDS. At last, all collected data are sent to BDS for monitoring and then data are resent to the upper computer for display. With the support of the combination of the techniques introduced above, the BDS-based approach can effectively and efficiently conduct SHM for structures.

The SHM objectives of this project in general can be separated into three parts—spatial positioning, displacement measurement, and stress measurement. These objectives are too complicated and professional to be performed using the traditional monitoring techniques, which depend on the facilities of the total stations, theodolites, and the manual field works. However, it will not be a problem for the Beidou system to achieve the objectives. 

For the spatial orientation, a Beidou satellite observation reference station is required adjacent to the site for the transmission of signals, as shown in Figure 14. At first, appropriate reference points are chosen to build local three-dimensional coordinates for determining monitoring locations based on the construction scheme via the BIM model that has been introduced in Section 2.3. Then, the space coordinates of the monitoring locations can be determined in the BIM model and transferred to the Beidou system to ensure that the components can be hoisted to the correct positions. The accuracy of the Beidou system’s positioning can be achieved as 1–3 cm, which completely meets the construction requirements.

The deformation monitoring, which includes displacement monitoring and stress monitoring, of high-altitude large-scale steel structures is known as one of the major difficulties in the area of structural health monitoring. The Tianhan system uses the Beidou local measuring system to compare the coordinates of the selected locations to monitor structural deformations. This monitoring can be automatically performed all day long without manual duty, and the accuracy can be within 3 mm, which also completely meets the requirements. Based on the simulation modeling introduced in Section 2.4, the monitoring locations can be found, and then the displacement and stress can be monitored via displacement and stress sensors which are connected to ZigBee sensing terminal nodes. The data can be transmitted using Beidou and Zigbee mobile local area network to the client acceptance platform to achieve real-time monitoring and display.

### 3.2. Displacement Monitoring

Displacement monitoring includes the component installation and temporary support unloading processes of the steel structure. Its main purpose is to prevent large displacement and angle of torsion by comparing the measured components of key structures with the calculated values and to ensure that the displacement of the structure meets the design indexes of the relevant specifications after the completion of construction. The obtained monitoring data can be compared with the warning values, and then the displacement of the structure under each working condition can be predicted to ensure construction safety.

In order to accurately monitor displacements of the monitored points, a scientific and reasonable monitoring control network is designed. The locations of the monitoring points are provided by the construction groups based on the datum points on site. Thus, geometrical morphology and geometric deformation of the structures can be fully reflected with the least control points, and displacements of all monitoring points can be measured during the whole construction procedure including the removal of temporary supports. During construction, the points with large displacement are mainly concentrated in the mid-span part of the truss, which requires strict and accurate monitoring. After the temporary support is removed, internal force redistribution will occur in the structure. Therefore, the displacements of the critical points, such as the mid-span part of the truss, need to be monitored to ensure the safety of the structure. The trusses of a single product are arranged according to 4 positioning and measuring points, as shown in Figure 15, and a total of 70 trusses of the project need to be monitored, which accounts for 280 monitoring points. 

### 3.3. Stress Monitoring

The reasonable layout of stress monitoring points for the temporary supports is essential safety work. During construction, the stress is mainly concentrated at the bottom of the temporary support columns which require 24-h monitoring to ensure the reliability of the temporary supports. In this study, the axial temporary supports (SY9TJ1 and SY10TJ2) are selected as examples for monitoring. The monitoring points are arranged as shown in Figure 16. SY9TJ1 and SY10TJ2 represent the number of stress monitoring points of temporary supports (the symbol of ■ indicates that a stress sensor is installed at one measuring point, and the length direction of the stress sensor is parallel to the axis of the component).

Corresponding stress monitoring is also needed for the main structure to ensure the correctness, accuracy, and safety of the construction process. According to the characteristics of the project and related engineering experience analysis, a single truss is arranged according to 8 positioning and measuring points, as shown in Figure 17. A total of 70 trusses of the project need to be monitored, which accounts for 560 monitoring points.

## 4. Results and Discussion

### 4.1. Typical Monitoring Results and Discussion

The Beidou system was used to conduct an all-around dynamic monitoring analysis on the selected truss in the construction process, and the data of the monitored truss can be obtained. The first trusses on both sides of Entrance 2, which are symmetrical and have identical geometries, were taken as examples of monitoring objects for analysis to show the typical curves. The geometry of Truss 1 is shown in Figure 18, which is as same as the ones shown in Figure 15 and Figure 17. Four displacement monitoring points were taken for each truss as shown in Figure 15. The monitoring results of the trusses (trusses 1 and 2) on both sides of the main entrance 2 are shown in Figure 19a,b, respectively.

According to Figure 19a, it can be seen that the maximum displacement occurs on the X-axis of the fourth monitoring point of truss 1, which is 20mm; while for truss 2, the maximum displacement occurs on the X-axis of the first monitoring point, which is 20.5 mm as shown in Figure 19b. The maximum displacements of truss 1 and 2 meet the requirement of the standard for the acceptance of construction quality of steel structures (GB50205-2020) [40]. The monitoring results could better reflect the safety and reliability of the field construction with the support of the simulated results provided by 3D3Sdesign2020 finite element software to ensure the construction process is safe and reliable. Thus, the BDS-Based approach can effectively monitor and evaluate the displacement-related structural health of the selected components.

Similar to displacement monitoring, the first trusses on both sides of Entrance 2 (truss 1 and truss 2) were taken as examples of monitoring objects for stress analysis. Eight stress monitoring points were taken for each truss as shown in Figure 17. Mid-span positions (points 5, 6, and 7 of truss 1 and 2, 5, and 8 of truss 2) were selected for monitoring. The stress monitoring results are shown in Figure 20. According to Figure 20, it can be seen that the maximum stress occurs on the sixth monitoring point of truss 1 and the second monitoring point of truss 2, which are 43.6 and 41.1 MPa, respectively. It can be found that the maximum stresses on the trusses are much less than the yield stress of Q355B steel (355 MPa) [39], and the stress-related structural health can be effectively monitored and ensured by the proposed BDS-based approach.

As introduced in Section 3.3, temporary supports of SY9TJ1 and SY10YJ2 were selected as monitoring objects for monitoring and data analysis. As introduced above, the temporary supports played an important role in the forming process of the MRSS, and hence the key nodes of SY9TJ1 and SY10YJ2 (places with large stress changes) were selected for analysis. As shown in Figure 21, the maximum stress is 38.1 MPa, which is nearly stable and far below the yield stress of Q355B steel. Thus, the temporary support and the steel structure are proven to be safe and reliable using the BDS-based approach.

### 4.2. Maximum Monitoring Results and Discussion

After 75 days of monitoring, the locations with the maximum displacement and stress were found, which were above the east and west entrances, respectively, as shown in Figure 22a,b. The maximum displacement and stress monitoring results are shown in Figure 23 and Figure 24, respectively. According to Figure 23, it can be seen that the maximum displacement is 36.2 mm in the X-axis, which meets the requirement of the Standard for acceptance of construction quality of steel structures (GB50205-2020) [40]. Similarly, it can be seen from Figure 24 that the largest stress is 56.9 MPa, which is far less than the allowable stress of Q355B steel. According to the monitoring results, it can be seen that the BDS-based approach can give evidence that the maximum displacement and stress of the structural component in the construction stage has a certain safety reserve and meets the requirements of the specifications and regulations. Therefore, the BDS-based approach proposed in this study can accurately and effectively complete the tasks of SHM for long-span steel structures in civil engineering.

### 4.3. Potential Applications, Limitations and Future Work

As a new member of the SHM approach community, the BDS-based approach can be applied in many engineering practical areas. For example, as GPS cannot provide reliable monitoring information for structural bridges due to obstructions, such as cables and passing vehicles [12], the application of the BDS-based approach can effectively deal with the satellite signal obstruction to achieve reliable SHM for bridges. Moreover, as GPS and GLONASS have been proven to be effective in SHM for steel structures [3], it can be predicted that SHM tasks of steel structures can be satisfactorily completed by the BDS-based approach. Recently, riverbed deformation monitoring for shield tunneling under rivers using GPS has been paid more attention to by researchers [41], so the BDS-based approach can also effectively conduct SHM in this area. Moreover, the BDS-based approach can be carried out in other SHM fields that have been applied by GPS, such as civil engineering, surveying, remote sensing, mechanical engineering, structural dynamics, signal processing, computational hardware and software, data telemetry, smart sensors, machine learning, and pattern recognition [41,42,43,44,45].

Although the BDS-based approach can effectively and efficiently carry out SHM in many engineering areas, there are still limitations to this approach. For instance, in the high latitude case, the precision of BDS deformation monitoring can be lower than those of low and middle latitude cases, which is mainly because of the small number of BeiDou satellites and the poor satellite geometry at higher latitudes in the experimental sites [5]. Fortunately, the integration of BDS and GPS measurements can improve the accuracy of BDS results significantly. The second limitation can be the requirement of other techniques, such as wireless techniques, numerical techniques, and BIM techniques. Moreover, the BDS-based approach needs to be applied combined with the Geographic Information System (GIS) technique to acquire more real-timely, accurate, and visual reflection in complex terrains [46].

For future work, more investigations and case studies in civil engineering are required to give more evidence and data for the BDS-based approach. Moreover, the BDS-based approach can be applied to other engineering areas, such as mechanical engineering, computational engineering, surveying, and mapping engineering. Last but not least, investigations of the combination of the BDS-based approach and other techniques, such as GPS, GIS, mathematical algorithms, BIM, and FEA, are also required in the future.

## 5. Conclusions

With the pursuit of architectural modeling by architects, more and more asymmetric and twisted large-scale spatial steel structures appear. The outer curtain wall of Tianhan Grand Theatre project is the large-scale spatial steel structure of the Mobius ring shape, and the structure gradually transitions from the initial circular axisymmetric or axisymmetric form to the asymmetric structure form. Thus, the distorted shape brings great difficulties to the construction and completion of the project. The installation process of the project is from the scattered unstable structure gradually transferred to a stable one, and structural positions are a gradual change process. Moreover, the intensive weld stress influence, position accuracy, and deepening molding difficulty require advanced technologies and management methods to ensure the completion of the project. Based on the above reasons, this project uses time-varying mechanics theory of construction, numerical simulation method and GNSS monitoring technology to build a Tianhan system to ensure the completion of the Mobius ring steel structure of Tianhan Grand Theater, and performed satisfactory achievements. By using the spatial structure design software 3D3SDesign2020, the numerical calculation model of the MRSS was established. The construction and unloading process of the structure were simulated and analyzed, and the steel structure’s inner ring trusses, rigid frames, and connections between the outer ring and the concrete structure were reinforced with the support frames. The internal force distribution of structural members in the installation process guides the safe implementation of the construction scheme of this project. With the help of BIM technique, the integral 3D model of Tianhan Grand Theater was built. Furthermore, with the combination of the total station and BIM three-dimensional model, the coordinate points of the steel structural model were determined based on the site lay-off point position. During assembly, after alignment with the origin, the position of each rod and node is controlled by the distance and coordinates, which can be monitored by Beidou System. After hoisting in place, measurement and correction are carried out according to the three-dimensional points of the Tekla Structure model to ensure the accuracy of the high-altitude installation, which can reduce adjustment and rework time, shorten the installation period, and reduce labor and material costs as much as possible. Besides the support of BIM and numerical calculation, a specific wireless communication technique is required for the transmission of signals. In consideration of this, the ZigBee technique is applied in this project to ensure the signals can be accurately and correctly transmitted.

With the support of BIM, numerical calculation, and ZigBee techniques, SHM can be implemented by BDS. At first, the specific Tianhan App was proposed and verified. Then the monitoring tasks, which in general can be separated into three parts—spatial positioning, displacement measurement, and stress measurement, were completed by the Tianhan App. The maximum displacement and stress of the MRSS were found by BDS after 75 dates of monitoring. It is found that the maximum displacement and stress of the structural component in the construction stage have a certain safety reserve and meet the requirements of the specifications and regulations. As a result, the application of BDS technology in combination with other supporting techniques provides a new idea for construction measurement and security control work, making the measurement of spatial positioning, displacement, and stress of spatial structure more reliable and convenient.

In short, this research project provides technical support for the construction work of Tianhan Grand Theater, makes a certain contribution to the smooth completion of the project, and also provides certain experience and references for similar projects. It can give a clear view to engineers and researchers that how to apply BDS in structural construction and provide a valuable real case for evaluating the performance of BDS in SHM.

## Figures and Tables

**Figure 1 sensors-23-05959-f001:**
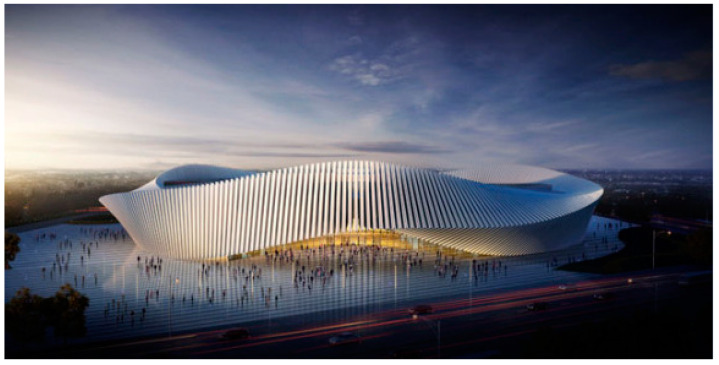
Predicted project rendering of Tianhan Grand Theater.

**Figure 2 sensors-23-05959-f002:**
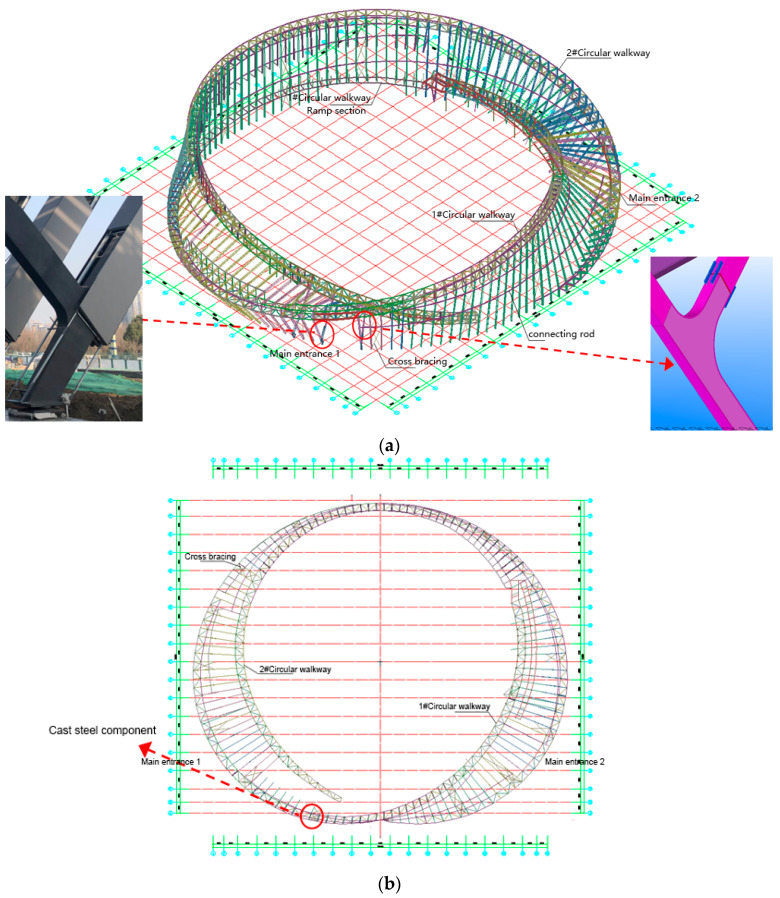
(**a**) Overview of the MRSS and details of the cast steel members; (**b**) Axonometric plan of the MRSS and locations of the cast steel members.

**Figure 3 sensors-23-05959-f003:**
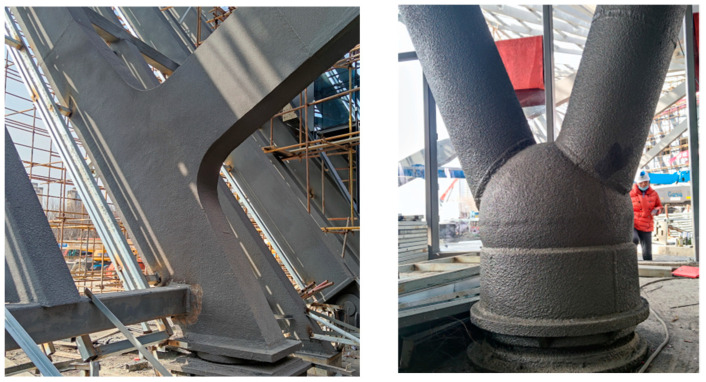
Examples of the complex joints.

**Figure 4 sensors-23-05959-f004:**
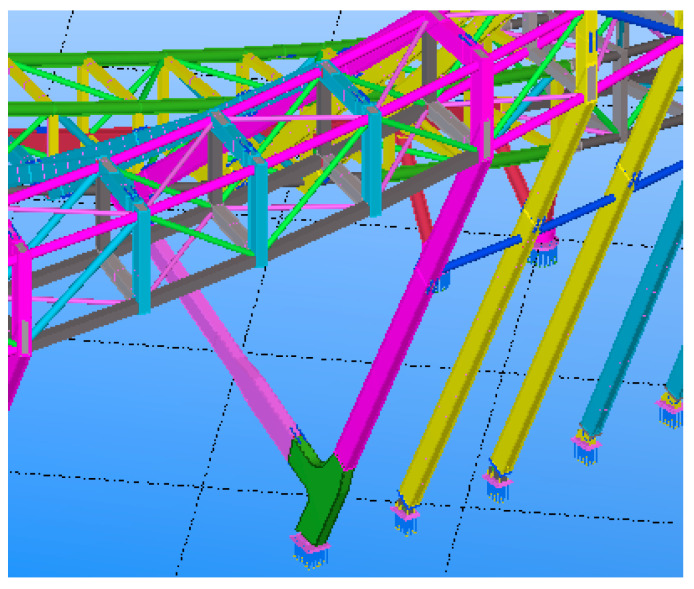
Tekla 3D model of outer curtain wall steel structure.

**Figure 5 sensors-23-05959-f005:**
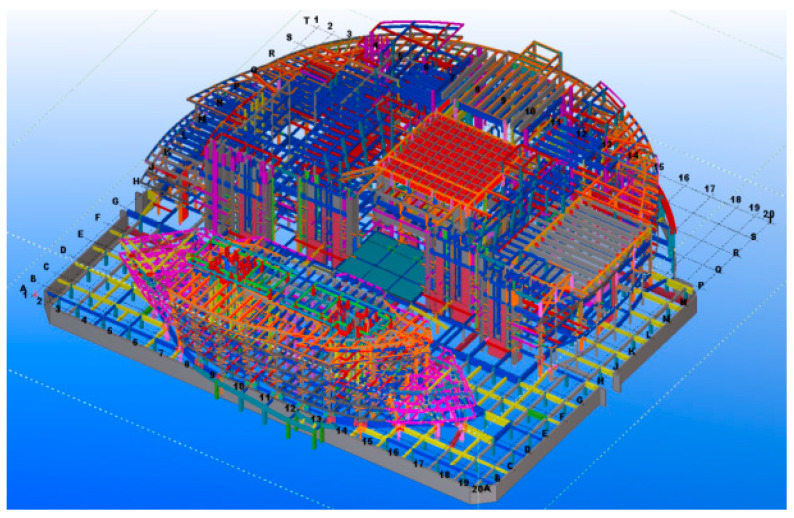
Tekla integral 3D model.

**Figure 6 sensors-23-05959-f006:**
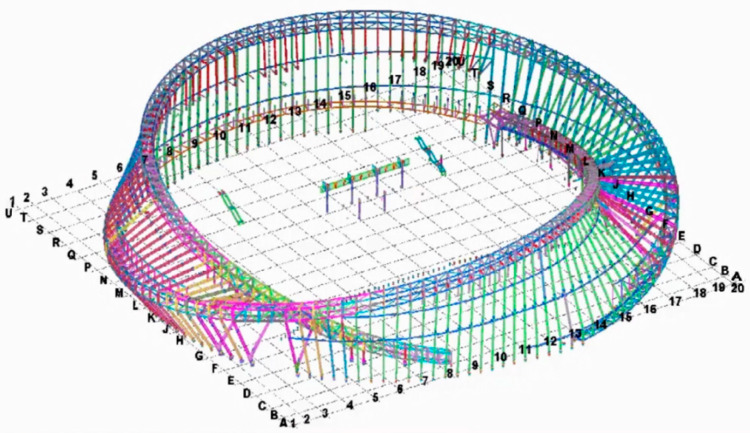
Component coordinate diagram.

**Figure 7 sensors-23-05959-f007:**
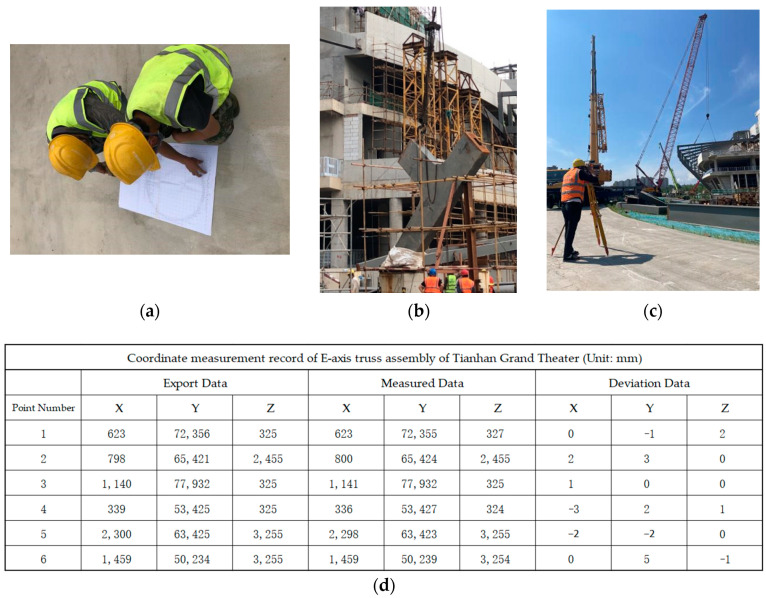
Using total station and BIM three-dimensional model to conduct onsite installation and positioning measurement. (**a**) Check the node coordinates; (**b**) Onsite lifting; (**c**) Field measurement correction; (**d**) Assemble data record comparison.

**Figure 8 sensors-23-05959-f008:**
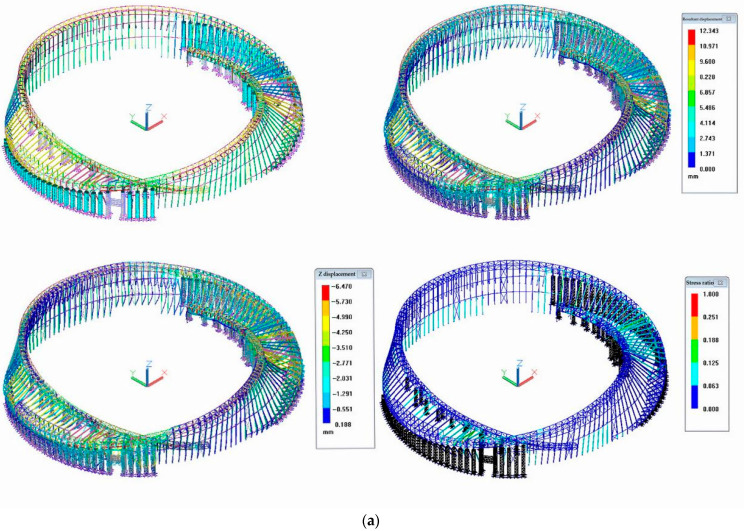
Stress and displacement simulation model: (**a**) The last step of lifting; (**b**) The last step of unloading.

**Figure 9 sensors-23-05959-f009:**
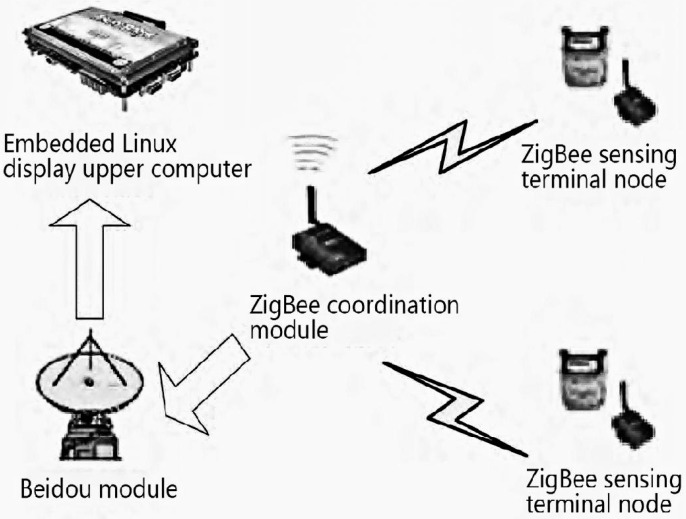
Overall architecture of the Tianhan System.

**Figure 10 sensors-23-05959-f010:**
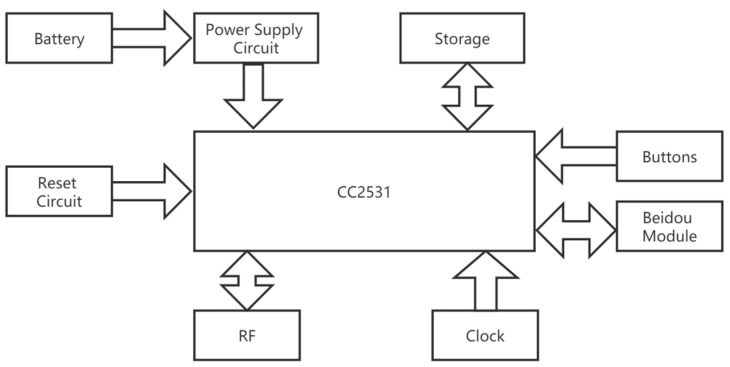
Hardware block diagram of ZigBee coordination module.

**Figure 11 sensors-23-05959-f011:**
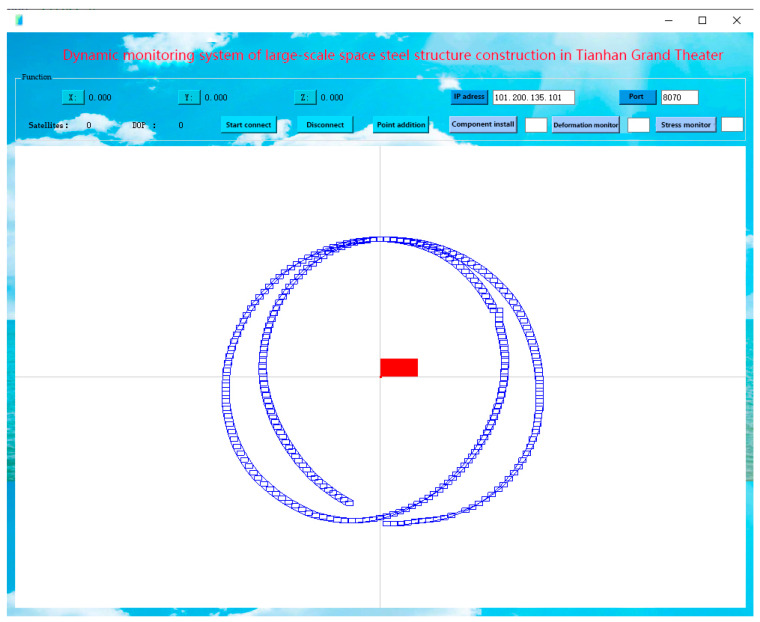
The main menu of the Tianhan App.

**Figure 12 sensors-23-05959-f012:**
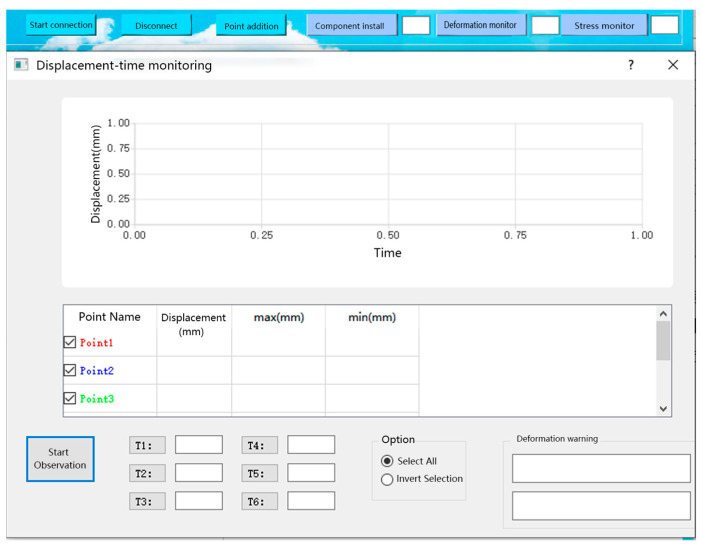
Displacement monitoring module.

**Figure 13 sensors-23-05959-f013:**
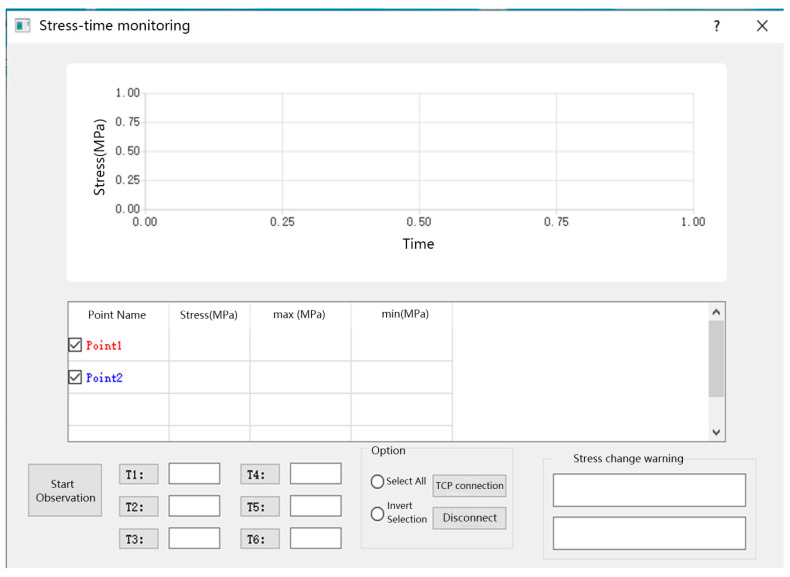
Stress monitoring module.

**Figure 14 sensors-23-05959-f014:**
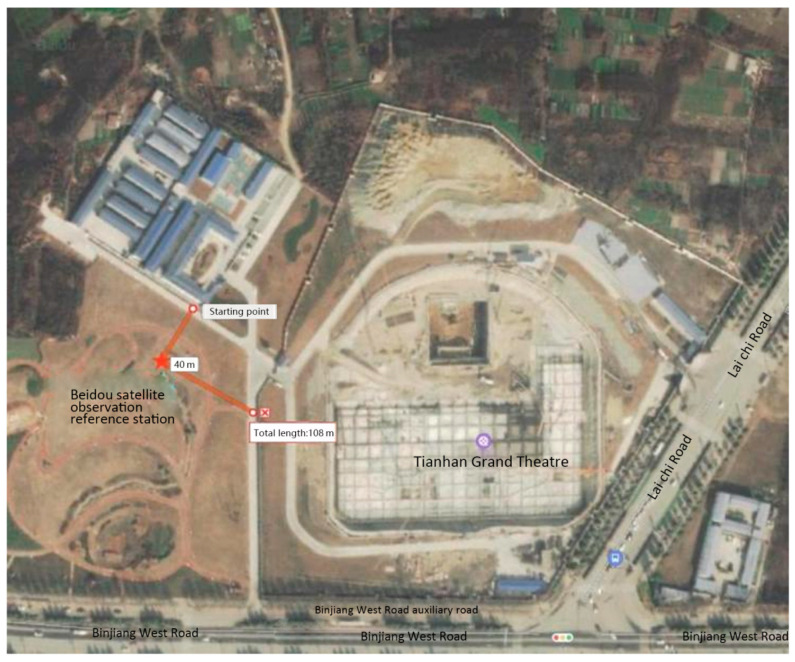
Location of the Beidou Satellite observation reference station.

**Figure 15 sensors-23-05959-f015:**
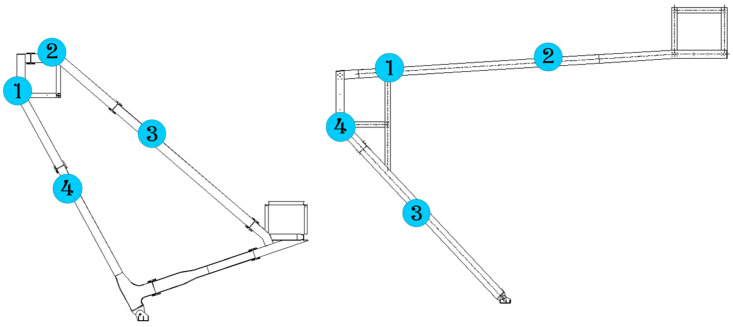
Arrangement of displacement measuring points.

**Figure 16 sensors-23-05959-f016:**
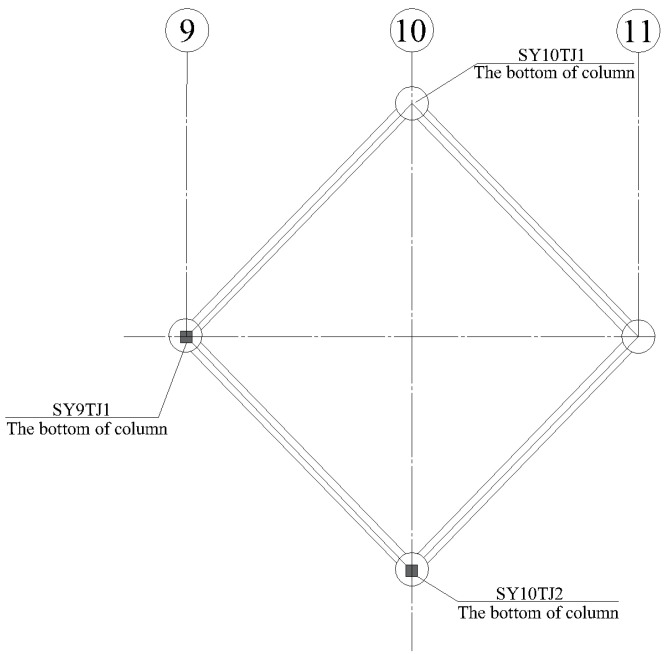
Layout of axial temporary support stress measuring points.

**Figure 17 sensors-23-05959-f017:**
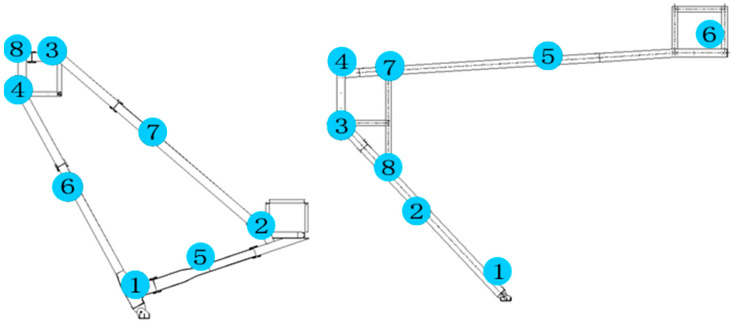
Arrangement of stress measuring points for the main structure.

**Figure 18 sensors-23-05959-f018:**
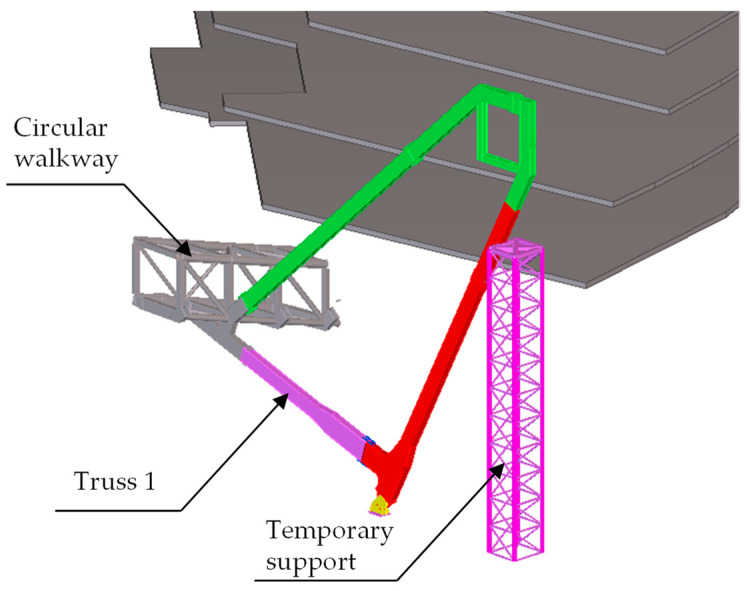
Truss 1 and the temporary support.

**Figure 19 sensors-23-05959-f019:**
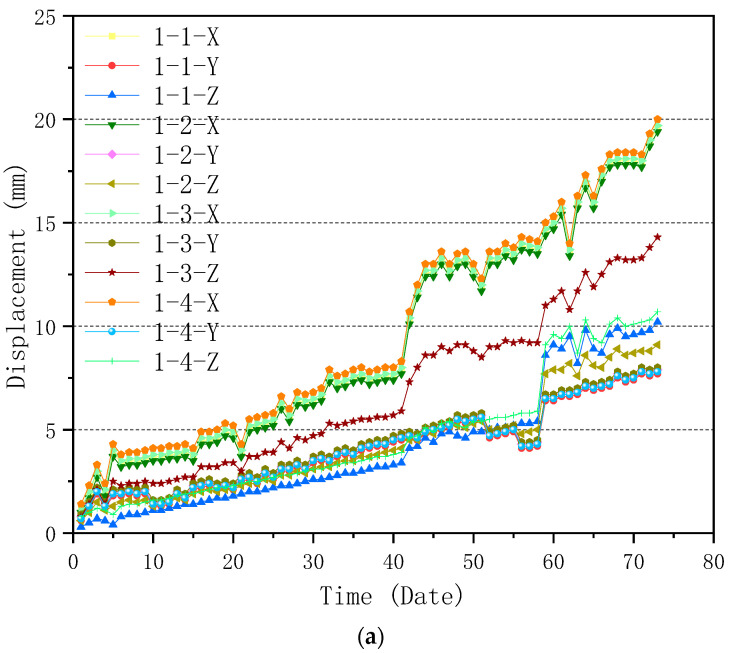
Displacement monitoring results of (**a**) Truss 1; and (**b**) Truss 2.

**Figure 20 sensors-23-05959-f020:**
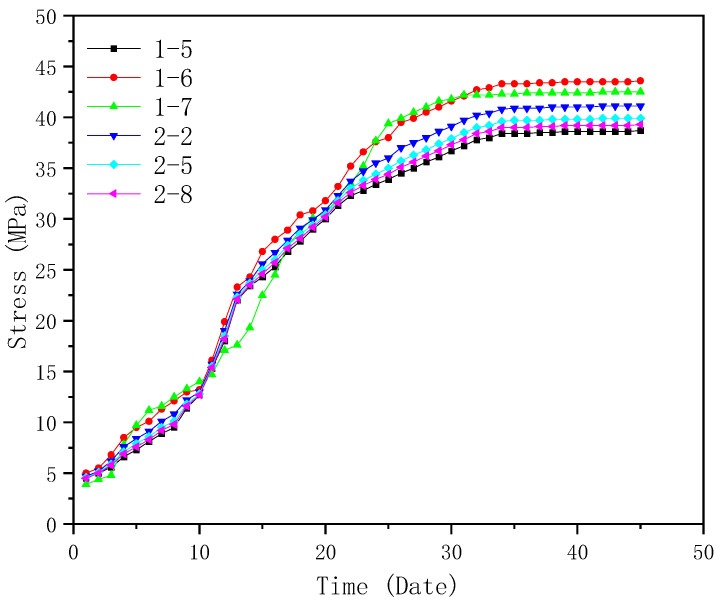
Stress monitoring results of Trusses 1 and 2.

**Figure 21 sensors-23-05959-f021:**
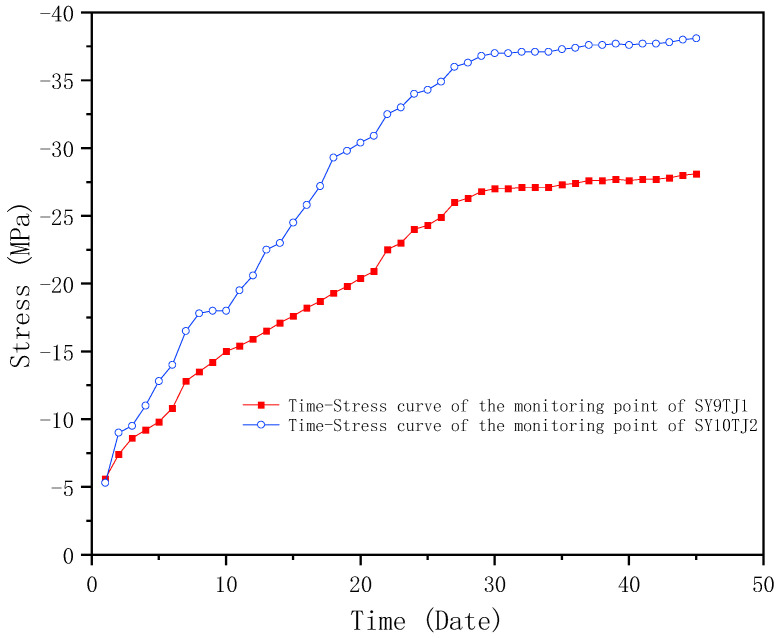
Diagram of typical temporary support stress variation.

**Figure 22 sensors-23-05959-f022:**
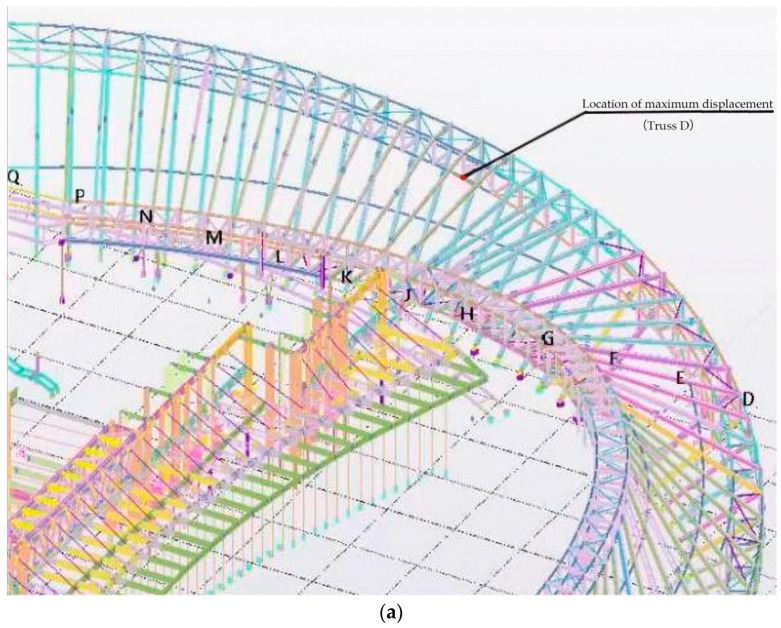
Locations of the largest displacement (**a**) and stress (**b**).

**Figure 23 sensors-23-05959-f023:**
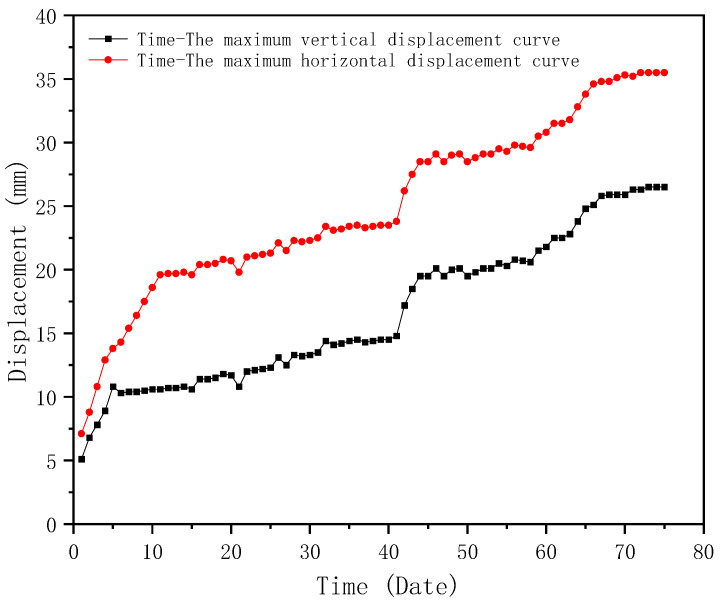
Displacement monitoring results of Truss D.

**Figure 24 sensors-23-05959-f024:**
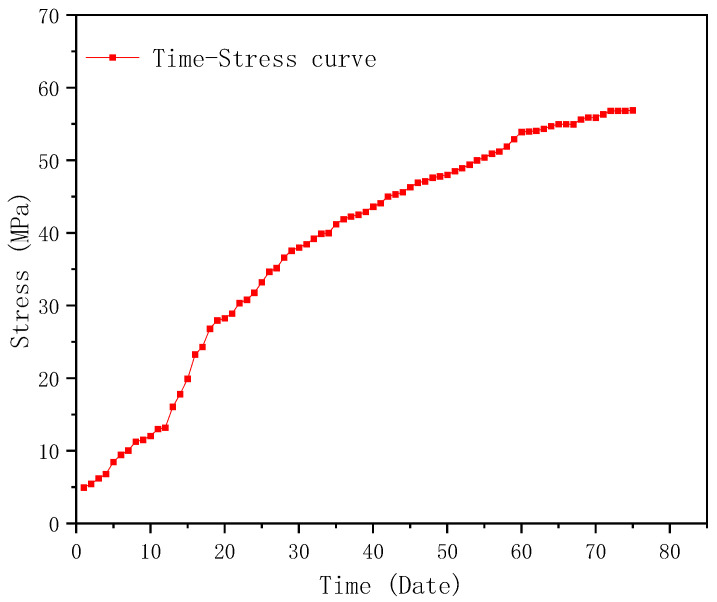
Displacement monitoring results of Truss S.

## Data Availability

Not applicable.

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
