# Peer review of "Structural Health Monitoring in Long-Span Steel Structures Based on the BeiDou Navigation Satellite System"

_sensors, 2023, doi:10.3390/s23135959_

Round 1

Author Response

Authors’ Response

Reviewer 1

Major comments: This paper reports an application of the BeiDou navigation satellite system in the structure health monitoring of a civil structure during construction stage. It also introduces the supporting hardware and software that are needed to apply the BDS. Through displacement monitoring and stress monitoring, the efficiency of the BDS was demonstrated. This represents an important alternative structure health monitoring technique for civil structures, especially for real-time, under construction structure measurement. Therefore, the subject of this paper is well within the scope of the Navigation and Positioning subsection of Sensor journal. The case study and results presented provide valuable information for the application of the BeiDou system and its accuracy in critical civil infrastructures. Therefore, I would recommend the acceptance of this paper by Sensor journal, provided that the following questions are addressed appropriately.

  1. The subject of this paper is focused on civil structures. So, I would recommend changing the title of the paper to reflect this point, and in the meantime, removing “with the support of simulation and communication techniques”.

Response: Thank you very much for the kind recommendations

The title of this study is changed to “Structural health monitoring in Long-span steel structures based on Beidou Navigation Satellite System” in the revised manuscript to reflect the research area of civil engineering, and “with the support of simulation and communication techniques” is also removed.

  1. Subsection 2.2 explained some challenges and complexity of structure health monitoring for the project and is not about some theories behind which. Therefor, I would recommend changing the title of this section.

Response: Thank you very much for the kind comments and recommendations

The title of Subsection 2.2 has been changed from “The theoretical basis” to “The BDS-based SHM system” in Lines 186 in the revised manuscript.

  1. For the stress monitoring, the paper mentioned a fiber optical strain gage was used. Later in section 2.5 it says that the stress measurement was conducted through the Zigbee sensing terminal node. This is confusing. I would recommend adding more information on the strain gage technique or the Zigbee sensing node technique.

Response: Thank you very much for the kind comments and recommendations

The authors are so sorry that the words “fiber optical strain gauges” make confusing, which have been revised to “displacement and stress sensors which are connected to ZigBee sensing terminal nodes” in Lines 390-391 in the revised manuscript. The Zigbee sensing terminal node technique is briefly introduced in Lines 245-247 in the revised manuscript. The description of “The ZigBee terminal node is responsible for connecting with the sensor device, receiving the analog signals of the sensor, and converting and transmitting digital signals to the ZigBee coordination node” can show more information on the ZigBee sensing node technique, which are in the Lines 299-301 in the revised manuscript.

  1. In line 67, it says ’BDS is providing better….’. This statement may be true but no detailed information is provided in the paper to support it. Therefore, I would recommend avoiding using such words like ‘better’ in this sentence and other similar sentences, unless the paper wants to present a comprehensive comparison between the current system and other navigation systems

Response: Thank you very much for the kind comments and recommendations

The word “better” is changed to “comparable” in Line 72 of the revised manuscript to avoid using such words.

The response is also attached as word file, please see the attachment.

Reviewer 2 Report

This study focuses on the application of the BeiDou navigation satellite system in structural health monitoring. By investigating its performance in the real project of Tianhan Grand Theater (TGT) and in conjunction with other techniques, the study demonstrates that BDS can effectively complete SHM tasks for long-span steel structures. Based on the authors' attainment of satisfactory results and the potential usefulness of the study to researchers, I recommend the publication of this study in Sensors, with the inclusion of the following comments.

1.      The abstract provided lacks a comprehensive summary of the work. It is crucial to include the significance of the study, research problem, methodology employed, main results obtained, and the future implications of the findings.

2.      The introduction of the current study lacks a concise statement regarding the research gap and objectives. It is essential to clearly articulate the research problem, identify the gap in existing literature, and highlight the novel contributions of the study.

3.      The references are lacking recent studies from the most recent years.

4.      The quality of the figures and pictures is not good, specifically figures 2.8 and 2.9. It is necessary to present clearer and higher quality figures and pictures.

5.      The study lacks a thorough discussion of the obtained results. There is a need to add a section (Results and Discussion) to discuss the implementation of the current study/results.

6.      What are some potential applications or practical implications of your findings for industry or society as a whole?

7.      Highlight some possible directions for future research based on your findings.

The English language proficiency is acceptable and meets the required standards.

Author Response

Authors’ Response

Reviewer 2

This study focuses on the application of the BeiDou navigation satellite system in structural health monitoring. By investigating its performance in the real project of Tianhan Grand Theater (TGT) and in conjunction with other techniques, the study demonstrates that BDS can effectively complete SHM tasks for long-span steel structures. Based on the authors' attainment of satisfactory results and the potential usefulness of the study to researchers, I recommend the publication of this study in Sensors, with the inclusion of the following comments.

  1. The abstract provided lacks a comprehensive summary of the work. It is crucial to include the significance of the study, research problem, methodology employed, main results obtained, and the future implications of the findings.

Response: Many thanks for the kind comments and suggestions

The abstract has been improved based on the reviewer’s suggestions. The improved abstract is shown in Lines 14-26 in the revised manuscript, and all revisions have been marked up using the “Track Changes” function.

  1. The introduction of the current study lacks a concise statement regarding the research gap and objectives. It is essential to clearly articulate the research problem, identify the gap in existing literature, and highlight the novel contributions of the study.

Response: Many thanks for the kind comments and suggestions

The introduction is significantly improved based on the reviewer’s comments in Lines 90-121 and 137-151, respectively, in the revised manuscript. The statement of research gap and objectives have been significantly improved by adding more literature and descriptions, and the contributions are also highlighted at the end of “Introduction”. All revisions have been marked up using the “Track Changes” function in the revised manuscript.

  1. The references are lacking recent studies from the most recent years.

Response: Many thanks for the kind suggestions

Around 10 recent SHM-related studies are updated in the revised manuscript in Lines 90-121 based on the reviewer’s suggestions, and the corresponding revisions in “Introduction” is also marked up using the “Track Changes” function in the revised manuscript.

  1. The quality of the figures and pictures is not good, specifically figures 2.8 and 2.9. It is necessary to present clearer and higher quality figures and pictures.

Response: Many thanks for the kind comments and suggestions

The quality of Figures 2.8 and 2.9 has been improved in Lines 276-280 and 310-311 in the revised manuscript, respectively, by presenting higher quality figures and improving the figure title. The corresponding description is also updated in Lines 266-267 in the revised manuscript.

  1. The study lacks a thorough discussion of the obtained results. There is a need to add a section (Results and Discussion) to discuss the implementation of the current study/results.

Response: Many thanks for the kind comments and suggestions

According to the reviewer’s comments, the title of Section 4 is changed from “Data Analysis” to “Results and Discussion” in Line 441 in the revised manuscript. Moreover, a new subsection of 4.3 is added to discuss the implementation of the current study/results and enrich the content of this section in Lines 517-547 in the revised manuscript. Besides, there are also many updates in Section 4 to improve the manuscript based on the reviewer’s comments and suggestions. All revisions have been marked up using the “Track Changes” function in the revised manuscript.

  1. What are some potential applications or practical implications of your findings for industry or society as a whole?

Response: Many thanks for the kind comments and suggestions

A new paragraph is added in Lines 518-531 in Subsection 4.3 in the revised manuscript to state the potential applications or practical implications of the findings for industry or society as a whole based on the reviewer’s comments. All revisions have been marked up using the “Track Changes” function in the revised manuscript.

  1. Highlight some possible directions for future research based on your findings.

Response: Many thanks for the kind comments and suggestions

A new paragraph is added in Lines 516-522 in the new subsection in the revised manuscript to state the possible directions for future research based on the reviewer’s comments. All revisions have been marked up using the “Track Changes” function in the revised manuscript.

The authors' response is also attached as the appendix, please see the attachment 

Reviewer 3 Report

The paper presents a case study on the implementation of a monitoring system for the construction of the Tianhan Grand Theatre. This system utilizes the Beidou GNSS system and ZigBee wireless communication technology. Its primary objective is to monitor deformations in critical positions within a large-scale steel truss structure, including displacement and stress monitoring. Overall, the paper is well-written and effectively structured. However, there are certain areas where improvements can be made to enhance its scientific merit and clarity:

1. While the topic is certainly important, incorporating more novel elements could significantly strengthen its contribution to the existing body of knowledge in the field. Exploring innovative ideas or approaches might elevate the overall impact and significance of the paper. Furthermore, the absence of comparative analyses or benchmarking against existing monitoring systems limits the credibility of the proposed approach.

2. The paper lacks a comprehensive discussion of the limitations associated with the implemented system, particularly in terms of accuracy, reliability, and robustness.

3. It is not evident how the strain gauge measurements are combined with the GNSS system. I encourage the authors to provide an illustration to clarify this aspect.

4. Please correct the position of the caption in Line 226.

5. In Section 2.4, it would be helpful to provide a brief description of the numerical model, including element types, loading protocols, and other relevant details.

6. In Line 404, please provide a reference for the yield stress of Q355B steel.

7. Please improve the quality of Fig. 2.6 and Fig. 4.5.

Author Response

Authors’ Response

Reviewer 3

The paper presents a case study on the implementation of a monitoring system for the construction of the Tianhan Grand Theatre. This system utilizes the Beidou GNSS system and ZigBee wireless communication technology. Its primary objective is to monitor deformations in critical positions within a large-scale steel truss structure, including displacement and stress monitoring. Overall, the paper is well-written and effectively structured. However, there are certain areas where improvements can be made to enhance its scientific merit and clarity:

  1. While the topic is certainly important, incorporating more novel elements could significantly strengthen its contribution to the existing body of knowledge in the field. Exploring innovative ideas or approaches might elevate the overall impact and significance of the paper. Furthermore, the absence of comparative analyses or benchmarking against existing monitoring systems limits the credibility of the proposed approach.

Response: Many thanks for the kind comments and suggestions.

A comparative analysis or benchmarking against existing monitoring systems is updated in the revised manuscript in Lines 90-122 to significantly strengthen the contribution to the existing body of knowledge in the field, which can elevate the overall impact and significance of this study. All corresponding revisions have been marked up using the “Track Changes” function in the “Introduction” of the revised manuscript.

  1. The paper lacks a comprehensive discussion of the limitations associated with the implemented system, particularly in terms of accuracy, reliability, and robustness.

Response: Many thanks for the kind comments and suggestions

A new paragraph is added in Lines 532-542 in the new subsection in the revised manuscript to state a comprehensive discussion of the limitations associated with the implemented system, particularly in terms of accuracy, reliability, and robustness based on the reviewer’s comments. All revisions have been marked up using the “Track Changes” function in the revised manuscript.

  1. It is not evident how the strain gauge measurements are combined with the GNSS system. I encourage the authors to provide an illustration to clarify this aspect.

Response: Thank you very much for the comments and recommendations

The authors are so sorry that the words “fiber optical strain gauges” make confusing, which have been revised to “displacement and stress sensors which are connected to ZigBee sensing terminal nodes” in Lines 390-391 in the revised manuscript. The Zigbee sensing terminal node technique is briefly introduced in Lines 245-247 in the revised manuscript. The description “The ZigBee sensing terminal node collects data from the sensor and transmits the data to the ZigBee coordination module through the ZigBee star network. Then, The ZigBee coordination module compressed-packages the data and appends the Beidou protocol. At last, the Beidou module transmits the data to the upper computer for real-time display” and “The ZigBee terminal node is responsible for connecting with the sensor device, receiving the analog signals of the sensor, and converting and transmitting digital signals to the ZigBee coordination node” can give an evident how the sensors are combined with the GNSS system, which are in the Lines 287-291 and 299-301 in the revised manuscript, respectively. All revisions have been marked up using the “Track Changes” function in the revised manuscript.

  1. Please correct the position of the caption in Line 226.

Response: Many thanks for the kind suggestions.

The position of the caption in Line 226 has been corrected in Line 280 in the revised manuscript. The revision has been marked up using the “Track Changes” function in the revised manuscript.

  1. In Section 2.4, it would be helpful to provide a brief description of the numerical model, including element types, loading protocols, and other relevant details.

Response: Many thanks for the kind suggestions.

A description of the numerical model is added in Lines 266-273 in Section 2.4 of the revised manuscript based on the reviewer’s comments. All revisions have been marked up using the “Track Changes” function in the revised manuscript.

  1. In Line 404, please provide a reference for the yield stress of Q355B steel.

Response: Many thanks for the kind suggestions.

A reference for the yield stress of Q355B steel is provided in Line 477 in the revised manuscript.

  1. Please improve the quality of Fig. 2.6 and Fig. 4.5.

Response: Many thanks for the kind comments and suggestions

The quality of Figures 2.6 “Component coordinate diagram” and 4.5 “Locations of the largest displacement (a) and stress (b)” have been improved in Lines 249-250 and 506-512, respectively,  in the revised manuscript.

The authors' response is also attached as the appendix, please see the attachment.

Reviewer 4 Report

The paper suggests that the BeiDou navigation satellite system (BDS) can be used in conjunction with other techniques for structural health monitoring (SHM) in real construction projects. Considering the case of the Tianhan Grand Theater, BDS data is investigated to identify potential structural issues.

1. The paper separately discussed several techniques: Building Information Modeling, ZigBee sensing, and theTianhan System.  However, the authors did not clearly show how these techniques are connected to produce displacement and stress monitoring results. Add a new subsection to discuss the combination of these techniques.

2. The authors showed several geometrical details of the structure and sense points, but it is not clearly mentioned that these details are for trusses 1 and 2, also whether trusses 1 and 2 have identical geometries.

3.  Improve the quality of the figure "Figure 4.5 Locations of the largest displacement (a) and stress (b)" and highlight Truss D  and Truss S.

4. Figure "Figure 4.1. Displacement monitoring results of Truss 1" and "Figure 4.2. Displacement monitoring results of Truss 2 " look identical, however in the text, it appears that there is a 0.5 displacement difference as the maximum displacement in the X axis. The authors discuss the close similarity in displacement results and justify this closeness. It is advised to show the difference in a figure to remove the confusion created by Figure 4.1 and 4.2. 

5. The authors need to enrich the introduction by discussing the recent SHM research such as "YUKI Algorithm and POD-RBF for Elastostatic and dynamic crack identification", "Damage Identification in Frame Structure Based on Inverse Analysis", "Experimental sensitivity analysis of sensor placement based on virtual springs and damage quantification in CFRP composite", "Vibration-Based Damage Assessment in Truss Structures Using Local Frequency Change Ratio Indicator Combined with Metaheuristic Optimization Algorithms"

Good language

Author Response

Authors’ Response

Reviewer 4

The paper suggests that the BeiDou navigation satellite system (BDS) can be used in conjunction with other techniques for structural health monitoring (SHM) in real construction projects. Considering the case of the Tianhan Grand Theater, BDS data is investigated to identify potential structural issues.

  1. The paper separately discussed several techniques: Building Information Modeling, ZigBee sensing, and theTianhan System. However, the authors did not clearly show how these techniques are connected to produce displacement and stress monitoring results. Add a new subsection to discuss the combination of these techniques.

Response: Many thanks for the kind comments and suggestions

A new paragraph is added to Subsection 3.1 in Lines 357-366 in the revised manuscript to clearly discuss the results of the combination of these techniques. All revisions are marked up using the “Track Changes” function in the revised manuscript.

  1. The authors showed several geometrical details of the structure and sense points, but it is not clearly mentioned that these details are for trusses 1 and 2, also whether trusses 1 and 2 have identical geometries.

Response: Many thanks for the kind comments and suggestions

Some descriptions are added in Subsection 4.1 (Lines 445-449) in the revised manuscript to clearly show that the geometrical details of the structure and sense points shown in Figures 3.2 and 3.4 are for Trusses 1 and 2. A new figure (Figure 4.1) is added in Lines 452-453 in this subsection to give more details about Trusses 1 and 2, and the related figure numbers are also updated. All revisions are marked up using the “Track Changes” function in the revised manuscript.

  1. Improve the quality of the figure "Figure 4.5 Locations of the largest displacement (a) and stress (b)" and highlight Truss D and Truss S.

Response: Many thanks for the kind comments and suggestions

The quality of Figure 4.5 “Locations of the largest displacement (a) and stress (b)” has been improved in Lines 506-512 in the revised manuscript. Truss D and Truss S are also highlighted in the improved figures and can be seen clearly.

  1. Figure "Figure 4.1. Displacement monitoring results of Truss 1" and "Figure 4.2. Displacement monitoring results of Truss 2 " look identical, however in the text, it appears that there is a 0.5 displacement difference as the maximum displacement in the X axis. The authors discuss the close similarity in displacement results and justify this closeness. It is advised to show the difference in a figure to remove the confusion created by Figure 4.1 and 4.2.

Response: Many thanks for the kind comments and suggestions

The original Figures 4.1 and 4.2 have been improved (in Lines 454-458 in the revised manuscript) by adding the dashed lines which can clearly show the differences between the y coordinates of the two figures. It can be seen that there is a 0.5 displacement difference as the maximum displacement on the y-axis. The numbers of the two figures are also updated to Figure 4.2(a) and Figure 4.2(b).

  1. The authors need to enrich the introduction by discussing the recent SHM research such as "YUKI Algorithm and POD-RBF for Elastostatic and dynamic crack identification", "Damage Identification in Frame Structure Based on Inverse Analysis", "Experimental sensitivity analysis of sensor placement based on virtual springs and damage quantification in CFRP composite", "Vibration-Based Damage Assessment in Truss Structures Using Local Frequency Change Ratio Indicator Combined with Metaheuristic Optimization Algorithms"

Response: Many thanks for the kind suggestions

The recent SHM research mentioned by the reviewer and some other literature are updated in the revised manuscript in Lines 90-121, and the corresponding revisions in “Introduction” is also marked up using the “Track Changes” function in the revised manuscript.

The authors' response is also attached as the appendix, please see the attachment.

Round 2

Reviewer 3 Report

The paper has been significantly improved, addressing the review comments, and is now suitable for publication.

Reviewer 4 Report

I have no further questions

good language